

# Effectiveness of gamification in nursing degree education

Sebastián Sanz-Martos[1], Cristina Álvarez-García[1], Carmen Álvarez-Nieto[1], Isabel M. López-Medina[1], María Dolores López-Franco[1], Maria E. Fernandez-Martinez[2] and Lucía Ortega-Donaire[1]

[1] Department of Nursing, University of Jaen, Jaen, Spain
[2] 061 Health Emergency Centre, Andalusian Health Service, Jaen, Spain

## ABSTRACT

**Background:** Previous research in nursing has found favorable results from the use of teaching methodologies alternative to lectures. One of the complementary methodologies used for university teaching is gamification, or the inclusion of game elements, creating a dynamic learning environment that allows the acquisition of knowledge and the development of other skills necessary for nursing students. The purpose of this study was to evaluate the effect of a gamification session on student satisfaction and knowledge scores in nursing students in simulated laboratory practice.

**Methods:** A pre-post quasi-experimental study was conducted with 122 students from the nursing degree program who participated in the research. The evaluation consisted of four sessions of three hours each. In two sessions, participants were trained through a vertical methodology, by means of theoretical training provided by teaching staff, and two hours of clinical simulation, where the students were able to practice the techniques on professional simulators. At the other two sessions, participants received an explanation of the theoretical aspects of the session, one hour of clinical simulation, and one hour of gamification, in which they had to pass tests based on the performance of practical activities on the subject of the session. At the end of the gamification session, all the participants received a certificate as winners of the "nursing game".

**Results:** There was an improvement in the satisfaction and the knowledge level in the gamification sessions. Both were statistically significant ($P < 0.001$). There was an improvement in the items related to the development of critical thinking and the mobilizing concepts from theory to practice in the gamification sessions.

**Conclusion:** The intervention was effective in improving the satisfaction of the sessions received and in knowledge development.

Corresponding author
Cristina Álvarez-García,
Cagarcia@ujaen.es

## INTRODUCTION

Currently, the main methodology used in university teaching is the master class, which can sometimes account for up to 70% of the total teaching hours. This methodology is considered an effective and fast method transmitting knowledge. In master classes, the responsible teachers play the main role, exposing the content to the students, who are

passive recipients of information; even so, master classes have limitations such as the loss of attention span over time. In recent years, new teaching methodologies have emerged that try to solve this limitation, such as peer education or flipped learning, learning through scenario simulation, gamification applied to teaching or game-based learning (*Gómez-Urquiza, 2019*; *Sadeghi, Sedaghat & Sha Ahmadi, 2014*; *Safari et al., 2006*; *Schmidt et al., 2010*).

University nursing teaching requires an innovative pedagogical approach to meet the increasing demands of an ever-changing healthcare environment. Gamification, which incorporates game elements into non-game environments such as educational contexts, has emerged as a potential strategy to improve undergraduate nursing students' academic engagement and performance (*Day-Black et al., 2015*).

The creation of a simulated scenario based on real cases or with simulated examples, but based on a health reality, accelerates the acquisition of technical skills, knowledge and abilities to manage problems that may arise when moving from theoretical content to real health situations (*LeBlanc, 2012*). In the case of biomedical sciences education, significantly greater benefits have been obtained in simulated scenarios than in traditional education based on vertical training (*Rohlfsen et al., 2020*).

There are positive results regarding the use of gamification in the training of health careers. The review by *Nieto-Escámez & Roldán-Tapia (2021)* evaluated the feasibility of incorporating gamification techniques for university teaching during the COVID-19 pandemic, finding high acceptance by students and obtaining improvements in academic performance. These improvements are associated with the high satisfaction reported by the participants, who showed high motivation to pass the games, as well as the presence of a favorable climate for meaningful learning. Another example of this is the review by *Martos-Cabrera et al. (2020)*, which found that university students, using games in non-playful environments such as teachers, mastered skills such as glycemic control through glycosylated hemoglobin; contributing with this innovative teaching methodology to an increase in the level of knowledge and positive attitudes toward healthy living habits.

In the specific context of nursing teaching, gamification has been associated with significant improvements in the acquisition of clinical skills, quick and accurate decision-making, as well as an increase in students' confidence and self-efficacy; in addition to the development of essential non-technical skills for nursing professionals such as effective communication, time management and problem-solving (*Day-Black et al., 2015*; *Gómez-Urquiza, 2019*).

Despite the benefits, it is crucial to address the challenges associated with implementing gamification in university education. Lack of technological resources, resistance to change, and the need for effective integration with the curriculum are key considerations that must be addressed to ensure the long-term success of this educational strategy (*Rodríguez-Torres et al., 2022*).

Therefore, gamification is presented as a promising tool to improve the training of university nursing students. The combination of game elements in a non-gaming environment, with theoretical and practical learning, can enhance the preparation of

future health professionals, providing them with the skills and confidence necessary to face the challenges of modern health care (*Jodoi et al., 2021*).

This is why the objective of this research was to evaluate the effect of a gamification session on student satisfaction and the knowledge scores of nursing students in simulated laboratory practices.

# MATERIALS AND METHODS

## Design

A quasi-experimental pre-post study was conducted to evaluate the satisfaction of nursing students and the scores obtained in the evaluation of the simulated laboratory practical sessions. This methodology was chosen in order to not deprive students of the best possible learning. The Strengthening the Reporting of Observational Studies in Epidemiology (STROBE) and Template for Intervention Description and Replication (TIDieR) guidelines were followed to report the study.

## Sample and settings

The target population was young students in the nursing degree program at a Spanish university during the 2021/2022 academic year. As inclusion criteria, we established students enrolled in the clinical nursing course who had not previously completed the laboratory practicum.

## Control and intervention about game-based learning

Participants were divided into 10 groups of approximately 12–15 people each and received two 3-h sessions about the necessary isolation measures depending on the type of patient to be treated and the techniques for removing foreign bodies in eyes and ears (without integrating any game-based learning aspect). These sessions consisted of a 1-h theoretical presentation by the lecturer in charge and two hours of clinical simulation in which the students carried out practice on prepared teaching simulators after visualization of the correct technique by the lecturer. The teacher corrected the students throughout the learning process to ensure they learned the techniques correctly. After these two sessions, the participants were asked to give feedback that included sociodemographic data, opinions and satisfaction regarding the teaching methodology and knowledge level.

The second part of the research consisted of exposing the same students to two 3-h sessions, using a teaching methodology based on game-based learning on the palpation and assessment of the lymph nodes and the neurosensory and vascular assessment of the feet of a patient with diabetes mellitus. These sessions consisted of a theoretical presentation by the teacher in charge for 1 h, then an hour of clinical simulation using the theoretical simulators, similar to the previous sessions, where the participants performed the techniques under the supervision and correction of the teacher. Finally, in the last hour of the session, the participants of the same students' group dressed as the participants of the television series "The Squid Game", that is, with green jumpsuits, which were indicative of being participants in the game, and the teacher in charge dressed in a red jumpsuit as the supervisor of the participants in that series. In this part, the participants had to

demonstrate their skills at various clinical stations independently. The first clinical station consisted of 15 clinical cases exposed in documents, in which the students randomly choose one each of these cases, and in front of the teacher, they would have to present the evaluation techniques of these patients with diabetic foot (previous history of ulcers, presence or absence of peripheral neuropathies, peripheral vascular diseases, possible presence of structural and biomechanical anomalies, and knowledge and attitude toward self-care on the part of the patient), as well as the identification measures of possible neuropathy (Semmes Weinstein monofilament, Rydel-Seifferf graduated tuning fork, Achilles reflex, thermal rod, and cotton swab or brush). The teacher recorded the corresponding measures or techniques the students named in each clinical case through a checklist The second station consisted of performing the exploration of a diabetic foot on a simulator, and assigning the correct nursing care. Each case was developed individually, with observation by the rest of the group, who were asked to note down potential areas for improvement, to be discussed at the end of the session. The teacher used a checklist for the interventions and activities proposed by the student in question; if interventions or nursing care were lacking, the student's final score for the activity, decreased in points. Finally, the third clinical station consisted of the assessment and palpation of the lymph nodes through 15 clinical cases; each clinical case was explained to a student in each round who acted as the patient, and a partner who was exposed to the test was the person evaluated; the realization of this station was also one by one. The professor used a checklist to verify that they had considered all the elements (node shape and size, sensitivity, mobility, and consistency). All students were subjected to the same level of difficulty in the tests. The technique developed was evaluated and not the result of lymph node localization, because there are some that are easier and others that are more complex.

The total score of the different checklists was used for the final qualification of the practice by the students. The students did not know the final mark of the practice until they completed the self-administered questionnaires (sociodemographic data, opinion, and satisfaction regarding teaching methodology and knowledge level) to avoid conditioning answer.

All students, received a certificate as winners of the "nursing game" (Fig. 1) for participating. The process of developing the sessions is shown in Fig. 2.

## Data collection

The data were collected by means of a questionnaire designed for this research composed of three blocks:

- Sociodemographic data and opinions: This block comprised five variables (sex, age, not having any previous experience with game-based learning, usefulness of the activity, description of the activity carried out). All variables had several precoded response categories, with an added option in which students could write another response. Only one response per variable was allowed except for the last variable where participants could indicate several options.

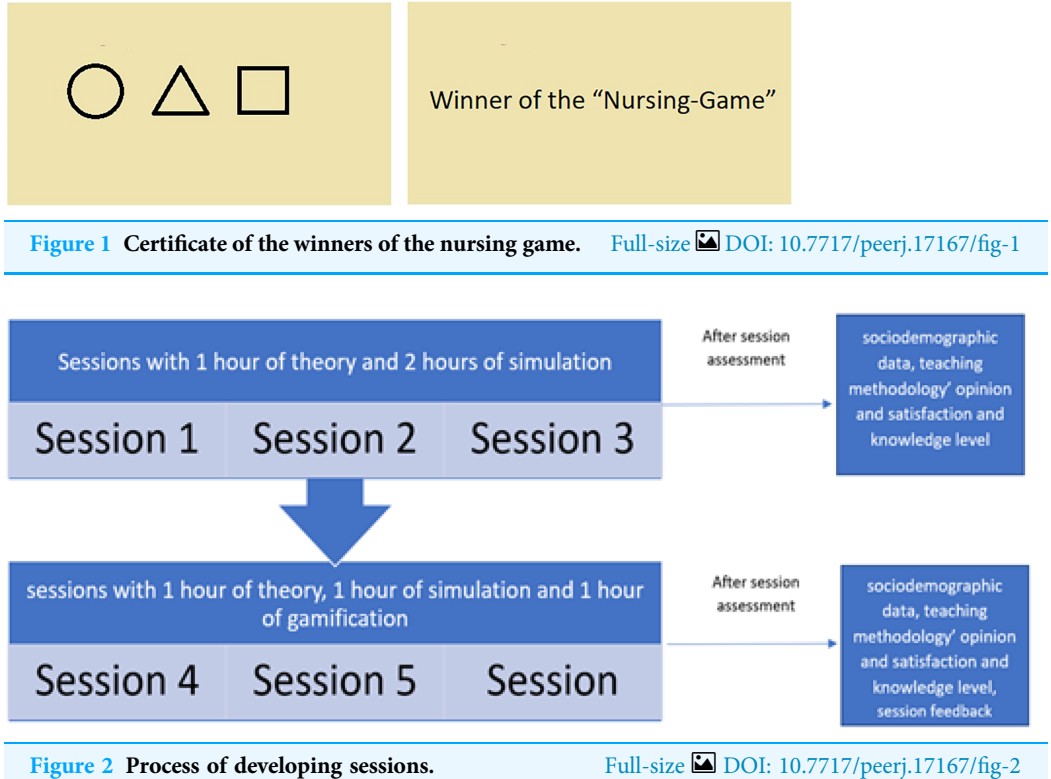

**Figure 1 Certificate of the winners of the nursing game.**

**Figure 2 Process of developing sessions.**

- Satisfaction feedback on the activity developed: We measured the feedback of the participants by means of a scale composed of 10 items measured on a Likert scale (1–5). The range of scores in the scale varies between 10 and 50 points maximum. The scale was made *ad-hoc* for this research and evaluated by a committee of national experts. The group consisted of four university lecturers of the nursing degree and three secondary school teachers in Spanish high schools. They were asked, using a Likert scale (1–5), for their evaluation of the relevance of the item for measuring the participants' satisfaction with a teaching activity and the clarity with which they had been written. The degree of agreement among the experts was evaluated by means of Aiken's V statistic, obtaining for the 10 items selected for the version used in the research a value of agreement higher than 0.7 points.
- Knowledge: It was evaluated by means of the questions of the course evaluation test, establishing as test variables 10 items on the course matter of the practical sessions in which the gamification methodology was used and as a test variable the rating of the 10 items of the theoretical-practical sessions in which a vertical methodology was used.

Participants were informed about the objective of this research and were asked to sign an informed consent after agreeing to participate in the study. During data collection, at least one member of the research team was present to answer any questions or problems related to the completion of the questionnaire. Completion of the questionnaire lasted approximately 10 min. Anonymity was guaranteed throughout the collection and processing of the information.

## Data analysis

The data were transcribed into an Excel document to begin the data analysis process. A descriptive analysis of sociodemographic information was carried out, obtaining their distributions in frequencies and percentages. The scores of the two scales (satisfaction and knowledge) were expressed by means of measures of central tendency and dispersion. The normality of the scores distributions was tested using the Kolmogorov-Smirnov test for the intervention and control scores on the feedback and knowledge variables. Both scales followed a non-parametric distribution.

A descriptive analysis was conducted of the feedback on the activity developed and knowledge items about the intervention and control sessions. The differences between the scores from the two types of sessions were calculated using the Wilcoxon test and the effect size using the Cliff Delta statistic. The level of significance was set at 0.05. All analyses were performed using SPSS version 27 and JASP version 0.17 for Windows.

## Ethical considerations

The Institutional Review Board of the University of Jaen approved this study (DIC.21/14. TFM). An information sheet was given to the participants. If students agreed to participate, they completed and signed informed consent prior to undertaking the session. Students were not obliged to participate and were reassured that this would not affect their progress and success in their course of study. Confidentiality of personal data was guaranteed. To compare the participants' information, they themselves chose their code, which was requested in the evaluation with the two teaching methodologies. No personal data or identifiers were included in the information collection tools. The entire procedure was collected and processing of the information was carried out in accordance with the data protection law of Spain (*Spanish Goverment, 2018*). Participants were informed and agreed to have their data published in a scientific article.

## RESULTS

### Descriptive analysis

The sample consisted of 122 students from a Spanish university's second year of the Nursing degree program. Most of the participants (79.5%) were females with a mean age of 20.77 years (SD: 5.03). Over half (61.5%) of participants did not know about the concept of gamification, and 75.4% had no previous experience with gamification (Table 1).

The main adjectives used by participants to describe the gamification experience were fun, focused during the game, and motivated. None of the participants felt bored during the gamification sessions, and all said that they would like to use it in future nursing degree classes. All the participants completed the game and received a badge as winners of the nursing game (Table 2).

### Satisfaction with the sessions

The satisfaction scale showed a high internal consistency value for the analysis sample, both in the evaluation of the vertical methodology ($\alpha = 0.893$) and for the sessions with gamification ($\alpha = 0.840$). The average score on the feedback on the activity developed with

**Table 1 Characteristics of the sample.**

| Variable | Categories | N (%) |
|---|---|---|
| Gender | Men | 25 (20.5%) |
| | Women | 97 (79.5%) |
| Previous experience with a gamification session | Yes | 30 (24.6%) |
| | No | 92 (75.4%) |

**Table 2 Results on the perception of the activity.**

| Variable | Categories | N (%) |
|---|---|---|
| Opinion about the gamification session | Fun | 106 (86.9%) |
| | Focus | 67 (54.9%) |
| | Tense | 52 (42.6%) |
| | Tired | 1 (0.8%) |
| | Bored | 0 (0%) |
| | Motivated | 89 (73%) |
| Future use in the nursing degree | Yes | 122 (100%) |
| Knowledge about gamification | Yes | 47 (38.5%) |
| | No | 75 (61.5%) |

**Table 3 Characteristics of the sample with both teaching methodologies.**

| Variables | Vertical methodology | Gamification methodology | P |
|---|---|---|---|
| Opinion on the activity developed* | 42.80 ± 6.38 | 47.98 ± 3.41 | Z = 8.229 |
| | | | P < 0.01 |
| Knowledge level* | 7.42 ± 2.15 | 8.68 ± 1.41 | Z = −5.776 |
| | | | P < 0.001 |

**Note:**
 * Data expressed as mean.

traditional methodology was 42.80 (SD: 6.38) points and 47.98 (SD: 3.41) points with the gamification methodology, improving 12.10%. This difference was statistically significant (Z = −8.229, $p < 0.001$; Delta: 0.567; Table 3).

There was an improvement in the score for all items with the gamification sessions, with significant differences in all of them, reaching a score of 4.90 points out of five points for item 2 "The exercises I have done promote my reasoning and critical skills" and item 5 "The activities have helped me to develop practical technical skills" about the perceived development of critical thinking and reasoning skills and practice (Table 4).

In the analysis of subgroups according to the dependent variables analyzed, we found a statistically significant difference in participants' satisfaction score for learning after the gamification session, with higher satisfaction among females. In the traditional methodology sessions, no differences were found for any of the participants' sociodemographic variables (Table 5).
**Table 4 Item analysis of the opinion scale.**

| Item | Session without gamification | | Sessions with gamification | | P |
|---|---|---|---|---|---|
| | M | SD | M | SD | |
| 1- I consider the information provided necessary | 4.35 | 0.91 | 4.75 | 0.54 | <0.01 |
| 2- The exercises I have done promote my reasoning and critical skills | 4.05 | 1.04 | 4.90 | 0.3 | <0.01 |
| 3- I feel prepared to act in a situation similar to the one presented. | 3.78 | 1.02 | 4.80 | 0.53 | <0.01 |
| 4- The activities carried out have helped me to make associations between the theoretical part and the practical part of the content. | 4.06 | 0.92 | 4.83 | 0.38 | <0.01 |
| 5- The activities have helped me to develop practical technical skills. | 4.30 | 0.95 | 4.90 | 0.43 | <0.01 |
| 6- I found the difficulty of the contents acceptable | 4.30 | 0.89 | 4.76 | 0.45 | <0.01 |
| 7- I found the resources proposed by the teacher to be sufficient. | 4.38 | 0.81 | 4.57 | 0.82 | 0.017 |
| 8- The material provided is easily understandable. | 4.53 | 0.79 | 4.85 | 0.54 | <0.01 |
| 9- The demonstrations by the teacher have helped me to learn how to carry out the techniques. | 4.48 | 0.82 | 4.75 | 0.65 | <0.01 |
| 10- I found the information clear and precise. | 4.57 | 0.76 | 4.87 | 0.50 | <0.01 |

**Table 5 Characteristics of the sample with the traditional and gamification methodology.**

| | Variables | Categories | Vertical | Contrast | Gamification | Contrast |
|---|---|---|---|---|---|---|
| Motivation | Gender | Male | 41.28 ± 6.22 | Z = −1.634 | 46.92 ± 5.17 | Z = −2.113* |
| | | Female | 43.19 ± 6.4 | | 48.26 ± 2.75 | |
| | Age | | | Rho = −0.129 | | Rho = −0.056 |
| | Have had any previous experience with gamification experiences | Yes | 41.70 ± 5.93 | Z = −1.511 | 47.50 ± 3.26 | Z = −1.218 |
| | | No | 43.15 ± 6.51 | | 48.14 ± 3.45 | |
| Knowledge | Gender | Male | 6.55 ± 2.53 | Z = −1.876 | 8.4 ± 1.67 | Z = −0.715 |
| | | Female | 7.64 ± 1.99 | | 8.75 ± 1.33 | |
| | Age | | | Rho = −0.088 | | Rho = 0.031 |
| | Have had any previous experience with gamification experiences | Yes | 7.63 ± 2.33 | Z = −0.997 | 8.83 ± 1.31 | Z = −0.642 |
| | | No | 7.35 ± 2.10 | | 8.63 ± 1.44 | |

**Note:**
* $P < 0.05$.

### Knowledge scale

For the knowledge scale, with traditional methodology, the score was 7.42 (SD: 2.15) points, and with the gamification methodology, 8.68 (SD: 1.41) points, improving 16.98% (statistically significant Z = −5.776, P < 0.001; Delta: 0.341; Table 3).

No statistically significant differences were found in the level of knowledge by any of the participants' sociodemographic variables (Table 5).

## DISCUSSION

This study aimed to evaluate the effect of a theoretical-practical session with gamification methodology on the modification of learning satisfaction and the level of theoretical knowledge acquired during the session. After the session, learning satisfaction increased by five points, on a maximum scale of 50 possible. An important factor to be studied in

learning environments is student satisfaction. It is considered an important outcome, and introducing gamification as part of the training process seems to provide a positive effect not only on satisfaction but also on motivation, both of which are important elements in student learning (*Ratinho & Martins, 2023*). The items that exhibited the highest percentage of satisfaction were those relating to the perceived effectiveness of developing critical thinking. The experience of *Jodoi et al. (2021)* found a similar result to our research, finding that participants exposed to a gamification session and using a mobile application significantly increased their rate of success in critical thinking. However, this result was similar to that obtained by the group that used the mobile application exclusively, so we must be cautious when interpreting the results. Although a striking result is the significant improvement compared to a control group in those that received training through a master class, highlighting the need to incorporate new teaching methodologies in university teaching. Gamification proved to be effective in mobilizing the theoretical concepts presented in the subject into a practical context and applied to reality, finding improvements in the development of healthy lifestyle habits (*Pérez-López, Rivera-García & Delgado-Fernández, 2017*), the development of empathetic attitudes toward people with functional diversity (*Oliver, Sterkenburg & Van Rensburg, 2019*) or the risk of infection by the human immunodeficiency virus (*Petersen, Beer & Dumbar-Kringe, 2011*). Gamification is an effective tool for visualizing processes; however, the modifications must be evaluated to ensure that they are maintained over time, as the effect of this intervention can be balanced and disappear. *Petersen, Beer & Dumbar-Kringe (2011)* states that we must be cautious when applying it to the modification of attitudes, as these may be influenced by other variables such as knowledge about the element or experiences prior to the evaluation, which has a significant effect on the creation of new attitudes, reinforcement of existing ones or modification of previous ones (*Rodríguez, 2004*).

The creation of a realistic learning climate is a fundamental element that allows nursing students to develop skills such as teamwork and working under pressure, essential skills to face the healthcare reality, where they will work in healthcare teams made up of people with different theorical knowledge and with a high level of care pressure. In our experience, and in a novel way compared to other studies, a competitive game was created where students had to pass each test in a certain timeframe, not have any penalty in the next test, and obtain the final prize as winner of the "game of nursing". In this game, they got exposed to pressure to perform tasks within the time available and perform them correctly. However, one aspect to highlight is that the teammates helped each other to continue playing, highlighting the development of camaraderie skills. During the game, teachers reported that they were able to help their classmates, seeking to create a favorable climate so that everyone could play and learn together. This result was similar in terms of the methodology used by researchers (*Gómez-Urquiza et al., 2019*), who exposed students to an escape room in which a group of five students had to collaborate to leave the room within a certain time limit under the supervision of teachers. In this case, the scenario kept them "locked in" to create pressure.

During the activity, our participants were focused, fun and motivated, reflecting previous findings in research on gamification, which described the experiences as

innovative, useful for learning and very satisfying, allowing the visualization of theoretical concepts that will be remembered (*Gómez-Urquiza et al., 2019*; *Pérez-López, Rivera-García & Delgado-Fernández, 2017*; *Sánchez-Martín et al., 2020*; *Tejero, 2021*).

The second variable of interest in our research, associated with modifying satisfaction with learning, was the level of knowledge developed through the gamification methodology. Our research found a statistically significant difference between theoretical knowledge of concepts addressed through gamification and concepts addressed through a traditional methodology. The gamification methodology proved to be effective in improving the level of knowledge, which was evaluated by scoring the items. This coincides with research by other authors such as *Azhari et al. (2019)*, *Ferriz-Valero et al. (2020)*, and *Luchi, Montrezor & Mancordes (2017)*. Previous studies, such as the one carried out by *Ezezika et al. (2018)* on the use of gamification to achieve knowledge about nutrition, showed similar results with a significant increase in the theoretical knowledge acquired that was linked to a greater motivation to learn and relate the everyday knowledge with new knowledge. In contrast, we find the study by *Nasiri et al. (2019)* that determines that games are insufficient for promoting learning results in nursing students.

Regarding the level of knowledge, one aspect to highlight is the gender differences found in previous research in which women obtained significantly higher scores than men (*Ferriz-Valero et al., 2020*). In our study, we found that the level of knowledge of women was higher than that of men, although it did not reach statistical significance. However, the satisfaction for learning did obtain a significant difference in favor of the women, who were more motivated to learn through gamification. This difference may be due to a greater predisposition to learning through mobile applications and social networks on the part of women, as well as their interest in the use of games as teaching tools, while men prefer games with a mainly recreational interest or application (*Holzmann et al., 2020*).

The incorporation of a teaching methodology related to a game is an element that significantly increased the level of satisfaction with the teaching received, as well as the level of knowledge, showing itself to be an effective and positive complementary methodology for teaching.

Among the main limitations of the research, it is worth highlighting that the contents taught through gamification and traditional methods were different. Therefore, we must be cautious in our interpretations, as the differences may be overestimated due to the different topics, and the differences in knowledge could be due to the content of the sessions and not to the teaching methodology used, which is a future proposal to be considered in new research. However, this is the most feasible way to conduct the study with existing resources, to compare traditional and new methodologies, and not to deprive and students of the gamification experience. On the other hand, satisfaction was measured after the gamification session and could be overestimated due to the high level of emotion associated with the game. For future research, we recommend assessing the development of practical skills as an outcome variable.

As a future line of research, the possibility of evaluating the usefulness perceived by the participants is proposed since, in our research we found a high level of usefulness and desire to use gamification in future sessions, but these two variables were not evaluated in

the sessions with traditional teaching methodology. The development of practical skills is another of the future areas to be measured, seeking to determine the effect both in terms of increasing the level of knowledge and the development of practical skills useful in the health professions. Finally, the participants' evaluation of their feelings during the gamification session should be addressed through qualitative methodology in future research.

## CONCLUSIONS

Students obtained an improvement in satisfaction and the level of theoretical knowledge after the gamification sessions, in addition to describing their experience with gamification as fun, with motivation and concentration during the game. All concluded that they would like to continue using this methodology in the rest of nursing degree classes. Using gamification as a complementary methodology in university teaching is an effective element for acquiring theoretical knowledge. Thus, it becomes clear that gamification should be used more consistently during the teaching of nursing degree courses.

## ACKNOWLEDGEMENTS

We would like to thank all the nursing students who voluntarily participated in this study.

### Funding

The authors received no funding for this work.

### Competing Interests

The authors declare that they have no competing interests.

### Author Contributions

- Sebastián Sanz-Martos conceived and designed the experiments, performed the experiments, analyzed the data, prepared figures and/or tables, authored or reviewed drafts of the article, and approved the final draft.
- Cristina Álvarez-García conceived and designed the experiments, performed the experiments, prepared figures and/or tables, authored or reviewed drafts of the article, and approved the final draft.
- Carmen Álvarez-Nieto conceived and designed the experiments, performed the experiments, authored or reviewed drafts of the article, and approved the final draft.
- Isabel M. López-Medina conceived and designed the experiments, performed the experiments, authored or reviewed drafts of the article, and approved the final draft.
- María Dolores López-Franco performed the experiments, authored or reviewed drafts of the article, and approved the final draft.
- Maria E. Fernandez-Martinez performed the experiments, authored or reviewed drafts of the article, and approved the final draft.

- Lucía Ortega-Donaire conceived and designed the experiments, performed the experiments, prepared figures and/or tables, authored or reviewed drafts of the article, and approved the final draft.

## Human Ethics

The following information was supplied relating to ethical approvals (*i.e.*, approving body and any reference numbers):

The University of Jaen granted Ethical approval to carry out the study (Ethical Application Ref: DIC.21/14.TFM)

## Data Availability

The data is available at Harvard Dataverse: Sanz Martos, Sebastián, 2023, "Replication blinder Data for Effectiveness of gamification in nursing degree education: A quasi-experimental study", https://doi.org/10.7910/DVN/KZ7600, Harvard Dataverse, V1.

## Supplemental Information

Supplemental information for this article can be found online at http://dx.doi.org/10.7717/peerj.17167#supplemental-information.

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
