# Peer review of "Effectiveness of gamification in nursing degree education"

_PeerJ, doi:10.7717/peerj.17167_

## Round 0.1 · original submission · Major Revisions

Dear authors,

Thank you for trusting PeerJ to publish your work.

After the peer review, we think that your article is not suitable to be published in its current version.

However, if you can address the reviewers' suggestions we could reconsider that decision.

Thanks and best wishes,

**Language Note:** PeerJ staff have identified that the English language needs to be improved. When you prepare your next revision, please either (i) have a colleague who is proficient in English and familiar with the subject matter review your manuscript, or (ii) contact a professional editing service to review your manuscript. PeerJ can provide language editing services - you can contact us at copyediting@peerj.com for pricing (be sure to provide your manuscript number and title). – PeerJ Staff

·

Basic reporting

Critical Review Report: "Effectiveness of Gamification in Nursing Degree Education (#93022)"
Clarity of the Problem Statement

The manuscript clearly presents the research problem. However, it would be beneficial to expand the contextualization of the problem, especially in relation to the current relevance of the topic in nursing education.
Methodology

The methodology appears appropriate for the proposed research. It is important to ensure that all methods are described in detail to allow for the study's replicability. Please clarify the methodology in detail.
Strength of Arguments

The arguments presented are well-structured. I recommend strengthening the discussion section by comparing the findings with previous studies in the field.
Relevance and Currentness of References

The references used are relevant and current. It would also be useful to include recent research that has employed similar methodologies to this study.
Quality of Writing and Structure

The writing is clear, and the document is well-structured. Some areas, such as the introduction, could be more concise.
Originality and Contribution to the Field of Study

The study contributes to understanding gamification in nursing education. It would be valuable to more clearly highlight how this work differs from previous studies.
Conclusions

The conclusions are consistent with the presented data. Discussing the practical implications of the findings more would be beneficial.
Recommendations for Improvement

Expand the contextualization of the problem in the introduction.
Provide more detail on the methodology used.
Compare the findings with previous studies in the discussion.
Include recent research with similar methodologies.
Make the introduction more concise and direct.
Highlight the originality of the study in relation to the existing literature.
Discuss the practical implications more in the conclusions.
Aspects to Review
Refinement of Language and Style

Improvement: Although the quality of English seems adequate, a comprehensive review by a native speaker or professional editor could improve the fluency and coherence of the language, eliminating possible grammatical or style errors.
Analysis and Discussion of Results

Improvement: The discussion should link the results with the existing literature, highlighting the study's contribution and relevance.
Explicit Ethical Considerations

Improvement: The study involves human participants, so it is essential to detail the ethical considerations, describing informed consent, ethics committee approval (date and name of the committee), and how personal data were handled. The legal reference (law supporting the data management procedure) should be indicated, specifically in Spain, this is regulated by the Organic Law 3/2018 (BOE nº294) of December 6, on Personal Data Protection and guarantee of digital rights.
Strengthening of the Introduction and Conclusion

Improvement: The introduction should clearly establish the context and importance of the study. The conclusions should highlight the key findings and their implications, emphasizing the study's contribution to existing knowledge.
Limitations and Future Prospects

Improvement: Include these sections.
Based on the above comments and recommendations, I suggest the authors address the highlighted aspects in this review report.

Best regards and happy new year

Experimental design

See main report

Validity of the findings

See main report

Reviewer 2 ·

Basic reporting

The English used is easy to read and to understand, the manuscript include relevant references, and the tables are easy to understand and include relevant data. The manuscript is an appropriate unit of publication.

Experimental design

The design is appropiate for the aims of the study and the methods show that the investigation has been performed following technical and ethical standards. I have some questions and recommendations to clarify some information.

Validity of the findings

All the data provided is easy to understand and robust based on the data collection and data analysis. Conclusions are linked to the data and the study aim.

Additional comments

The manuscript is well written, easy to read and to understand and I think that the topic is quite interesting and relevant for university education.
I have some recommendations.
INTRODUCTION
- I would include in the definition of gamification that "involves the use of game elements in non-game environments"
METHODS
- Did you established any exclusion criteria? If so, please specify it.
- I think that it is necessary to include a subsection in the methods section named: "control and gamification intervention" that includes the information of both the control and the gamification intervention.
- Including a figure to simplify the explanation of the different intervention/sessions and hours would be of interest for the readers.
- Did the participants receive any type of points or penalties during the game? Or just the final evaluation grade?
RESULTS
- What was the maximum score for satisfaction?

Kind regards

·

Basic reporting

Thank you for doing this work on gamification. There are numerous areas in your paper with English errors regarding tense, length of sentences, and punctuation. Your abstract needs more clarification. See the following paragraph.
It is not clear if all students attendant all four sessions. Line 23 says you had 4-3 hour sessions however in line 24 it says they were trained with vertical methodology and one hour of clinical simulation please define if the vertical methodology took two hours. It is not mentioned if the first two sessions received any evaluations. Also in line 24 you should end this sentence after simulation and begin a new sentence. Again it is not clear in the second two sessions that it took an hour to explain the theoretical aspects of this session since you only define what two hours were spent on. In your results section it is unclear what the development of critical thinking in mobilizing concepts means or how they were measured, since you already said they improved satisfaction and knowledge. In your conclusion, the knowledge they gained is only about the content they learned not overall knowledge development about gamification.
In line 59, you might want to add pure evaluation along with your discussion of pure education. I suggest you move line 84 2 lines 76 to begin this content of literature evidence. Line 88 is very long and needs to be rewritten. The evidence you discussed presents clinical use of gamification and not gamification in academia. It would be best to show examples from academia rather than clinical uses.

Experimental design

line 101 and 102 You should explain the Strobe and the Tidier guidelines. You need to state why you used these guidelines and what did they measure, was it for assessing the knowledge or the satisfaction? There is no statement of hypotheses for this quasi experimental study. It is unclear why you had different content of education for these groups. Isolation and eye care are different then palpating and assessment of lymph nodes and neuropathies and diabetes. This is not the best design for a study 3 The best design would have been randomized control and experimental. Both groups receiving the same education with the experimental also getting the gamification treatment. Please explain why you used different type of educational content to compare to gamification content.
Line 129 reviews the fact that you gave 15 clinical cases and the students had to choose from one of these cases, this leaves a lot of variability and should have been standardized between all the students, not leaving it up to this tune to choose which case study they wanted. You need to clarify what the final score being decreased means in line 140. Also in line 140 there should have been standard assessment for certain lymph nodes rather than allowing the student to choose which lymph nodes they would assess. In line 145 you state that all students are subjected to the same level of difficulty in the tests. However, it is more difficult to find certain lymph nodes than other lymph nodes and that was not taken into consideration. In line 157 define what previous experience you were looking for, was this clinical experience?
It is good that you validated your feedback scale. At the end of data collection you should discuss how the data from the paper forms were entered into your statistical software, assuming you used paper to collect your data.

Validity of the findings

In line 183 please name or define which two scales you are talking about, satisfaction and knowledge? Move sentence 198 two before the sentence in 197 and combine the content 3.
Inline 204 please explain “no previous experience”- are you discussing academic gamification or games in general- as your statement is not clear.
Were the students offered to to give feedback to the sessions without gamification- inline 207 new state they did not feel bored during gamification however you do not state if they felt bored during the previous education without gamification, again all of these evaluations should be given to both groups.
Was the satisfaction scale in line 212 a standardized scale or was this scale also assessed for validity by experts?
Why is line 211 listed as a modification? Line 225 needs explanation of what a deeper analysis means, is this the modification?
Line 227 states there was number differences for the socio demographic variables in the tradition method session however all the groups were the same so shouldn't this be the variables for all of the students? This concept of difference in variables needs to be clarified. In line 245 you mentioned results that improved critical thinking, however you are not assessing for critical thinking - only satisfaction and knowledge. The sentence that begins in line 262 is very long and needs editing. In line 276- you discuss your study so it is confusing why you have three other articles cited when only discussing your study? Define what you mean in line 286 regarding “hit rate”, it is unclear how this fits your study. Why wasn't satisfaction measured after the non-gamification session?

Additional comments

It is good to have completed a study on gamification. However it is not clear why you chose this design and you really need to state up front why you chose it. You do discuss that it is a limitation at the end of your paper however you will confuse the reader by not discussing it up front. This is a critical problem in this paper. Will need editing on English and grammar.

---

## Round 0.2 · accepted · Accept

Authors have modified the paper according the reviewers' issues so I think that the manuscript is ready for publication.

Reviewer 2 ·

Basic reporting

Thank you for addressing my comments and questions

Experimental design

Thank you for addressing my comments and questions

Validity of the findings

Thank you for addressing my comments and questions

Additional comments

Thank you for addressing my comments and questions

·

Basic reporting

no comment

Experimental design

Weak design for gamification - not clear as to how being in a costume makes it a gamification study. Sustainability with the squid game may not work going forward.

Validity of the findings

no comment

Additional comments

glad the authors pointed and explained the different topics of education used as a caution - starting with line 308- thank you.